# Defining No and Low (NoLo) Alcohol Products

**DOI:** 10.3390/nu14183873

**Published:** 2022-09-19

**Authors:** Alex O. Okaru, Dirk W. Lachenmeier

**Affiliations:** 1Faculty of Health Sciences, University of Nairobi, Nairobi P.O. Box 19676-00202, Kenya; 2Chemisches und Veterinäruntersuchungsamt (CVUA) Karlsruhe, Weissenburger Strasse 3, 76187 Karlsruhe, Germany

**Keywords:** no-alcohol products, lower-strength alcohol products, substitution, alcohol harm, artisanal production

## Abstract

Reducing the alcoholic strength in beverages as a strategy to reduce harmful alcohol use has been proposed by multilateral institutions such as the World Health Organization and governments worldwide. Different industrial and artisanal techniques are used to achieve low-alcohol content beverages. Therefore, regulations regarding the content of alcohol in beverages and strategies to monitor compliance are important, because they are the main reason for classification of the beverages and are central to their categorization and market labelling. Furthermore, analytical techniques with adequate sensitivity as low as 0.04% vol are necessary to determine the alcohol ranges necessary for classification. In this narrative review, the definitions of no and low (NoLo) alcohol products are described and the differences in the legal definitions of these products in several regions of the world are highlighted. Currently, there is clearly confusion regarding the terminology of “no”, “free”, “zero”, “low”, “light”, or “reduced” alcohol products. There is an urgent need for global harmonization (e.g., at the Codex Alimentarius level) of the definitions from a commercial perspective and also to have common nomenclature for science and for consumer information.

## 1. Introduction

Recently, popularity and consumer demand for low-alcohol and alcohol-free versions of beverages such as beer, wine, spirits, and cocktails has increased [1,2]. In a period of increasing awareness of alcohol-related health risks, the emphasis has been placed on beverages that help to protect and improve health conditions. Numerous acute and long-term harmful effects of alcohol use have been reported in the literature. Acute effects of alcohol consumption include, in particular, the noticeable impact on the central nervous system (CNS) including drowsiness, slurred speech, poor judgment, irrational behaviour, euphoria, and reduced sensory and motor abilities [3]. Confusion, stupor, coma, and ultimately death are the result of sustained elevation of ethanol concentrations in the CNS. Other acute effects are gastric mucosa irritation and suppression of myocardial contractility, as well as inhibition of antidiuretic hormone function, leading to increased diuresis and accompanying dehydration. Among the long-term effects of alcohol are liver damage, cancers, alcohol dependence, loss of memory, dyslipidaemia, cardiomyopathy, and iron and folate deficiency anaemias [3,4,5]. Taking into consideration these potential health risks of alcohol consumption, an acceptable daily intake (ADI) value of 2.6 g/day has been recommended [6], and low-risk drinking guidelines are available [7].

As a result, a group of alcoholic beverages commonly referred to as no or low (NoLo) alcohol products has emerged. NoLo alcohol products are beverages (such as beer, spirits, wine, and cocktails) that normally contain ethanol as an ingredient but are produced with ethanol completely removed or significantly reduced. In the market, there is a broad range of terminology that refers to NoLo, and the most common examples of terms range from “no”, “free”, “zero”, “low”, “light” to “reduced”. The operational definitions of the terms “alcoholic beverage”, “low-alcoholic beverage” (LAB), and “non-alcoholic beverage” (NAB) are subject to varying regulations in different countries. According to the EU Regulation 1169/2011 on the provision of food information to consumers, beverages with “alcoholic strength by volume” greater than 1.2% vol. are called “alcoholic beverages”, but there is no implicit EU regulation for the range below 1.2% vol. [8]. For most European countries, the limit for a “no-alcohol beverage” is considered to be 0.5% vol. [9,10]; however, there is often not a clear rationale provided for this limit. On the one hand, this limit may have been related to the levels of alcohol tolerated in other types of fermented foods and beverages such as fruit juices, bread, vinegar, or kefir; on the other hand, it may have been related to the technological possibilities to manufacture alcohol-free beer in the 1970s when the first products were developed (which typically were at 0.4% vol., and therefore, the limit may have been pragmatically set at 0.5% vol.). A final rationale may have been toxicological or biochemical considerations that less than 0.5% vol. may not cause psychoactive effects or blood alcohol levels, which would impair driving abilities, even following excessive drinking [11]. Hence, such a level of alcohol would be seen as a virtually safe dose.

Nevertheless, there is a patchwork of different practices around the world. For example, in Germany, a beverage is considered to be alcohol free if the maximum alcoholic strength is below 0.5% vol., while Spain is at variance with this, setting a maximum limit of 1% vol. for no-alcohol products. Moreover, in France, alcohol-free beers contain a maximum of 1.2% vol. [12]. In the United Kingdom, alcohol-free beverages have a maximum of 0.05% vol., which is ten times less than their EU counterparts. In the USA, non-alcoholic beverages should contain no more than 0.5% vol. In Asiatic countries, namely China and Japan, the limits of ethanol in non-alcoholic beverages are set to be not more than 0.5% vol. and 1% vol., respectively [12]. Clearly, it can be seen that the regulatory limits for delineating NoLo alcohol products are divergent even for the same geographical or cultural regions.

In this review, the legal definitions of NoLo alcohol products in different regions around the world are examined with regard to alcoholic strength and briefly the quality control aspects of the products are highlighted. The classification of NoLo alcohol products is also proposed to include artisanal and alcohol-flavoured non-alcoholic beverages. 

## 2. What Actually Is “Alcohol” and an “Alcoholic Beverage”?

Before focusing on the definition of NoLo alcohol products, briefly, the components of an alcoholic beverage are considered. Chemically, the term alcohol refers to any organic molecule that contains a hydroxyl group (-OH) covalently bonded to the carbon backbone of the molecule. The simplest aliphatic alcohol is methanol (H_3_C-OH), followed by ethanol (H_3_C-CH_2_-OH), and several, so-called higher alcohols with longer chain lengths, and also branched chains of more than two carbon atoms. Higher alcohols are also sometimes called “fusel alcohols”. This group of compounds may also contain non-aliphatic alcohols such as phenylethanol. 

In alcoholic beverages produced by natural fermentation, such as wine, beer, and spirits, there is a mixture of all these alcohols that are subject to wide variations, ranging from vodka, which is almost pure ethanol and contains only traces of methanol and higher alcohols, to fruit or fruit marc spirits, which may contain considerable levels of these compounds, sometimes exceeding 1000 g/hL of pure alcohol. However, ethanol is, by far, the dominating compound (typically > 99% of all alcohols), and therefore, ethanol concentration and alcoholic strength are often used synonymously. When people start thinking about NoLo alcohol products, they are believed to primarily consider reductions in ethanol and not in the other alcohols.

Interestingly, the EU food information Regulation 1169/2011 [8] specifies that the “actual alcoholic strength by volume” shall be indicated; therefore, the regulation is not specific to ethanol. The Organisation Internationale de la Vigne et du Vin (OIV) is more specific in its definition of alcoholic strength and notes that “homologues of ethanol, together with the ethanol and esters of ethanol homologues are included in the alcoholic strength” [13]. The inclusion of homologues can be explained by the simple fact that all reference methodologies for determining alcoholic strength are based on indirect methods (typically distillation and measurement of density), and therefore, ethanol is not specifically analysed.

In conclusion, it should be stressed that the “alcoholic strength” of an alcoholic beverage is a term based on analytical convention methods that include ethanol, methanol, higher alcohols, and other volatile compounds that are determined using reference methods based on densimetry.

When referring to NoLo alcohol products in the following discussion, it must be considered that the nomenclature is sometimes unclear and often vague if the authors refer to “alcoholic strength” or more specifically to “ethanol concentration”. Typically, the authors believe that even when studies refer to “ethanol”, they can actually mean “alcoholic strength” as there is no internationally established standard to specifically measure ethanol in alcoholic beverages.

In the EU, the correct nomenclature for alcoholic strength according to the EU food information Regulation 1169/2011 [8] is that it should be indicated by a number to not more than one decimal place, followed by the symbol “% vol.” and may be preceded by the word “alcohol” or the abbreviation “alc”. In this article, this nomenclature is used and alcoholic strength is stated with the unit “% vol.”.

It is the authors’ belief that the concept and definitions of the various no and low alcoholic beverages should be similarly based on the EU concept of alcoholic strength (which includes homologues) rather than ethanol alone.

## 3. Differences in the Definitions of No and Low (NoLo) Alcohol Products

Table 1 compares definitions of “alcohol free” (no alcohol) and, where applicable, low-strength (low-alcohol) limits used in different countries, and shows that there is considerable variation among countries, that is, there is no standard definition even among EU member states and there are considerable differences among countries’ predominant mode of market regulations. It can also be observed that most countries identify alcohol-free (no-alcohol) products as being below an alcoholic strenght of 0.5% vol. The UK, previously in the EU, has imposed stricter limits on what is deemed to be alcohol free, which can have implications for trade and consumer expectations of what really is “alcohol free” (Table 1). 

As shown in Table 1, some countries have not set limits applicable for NoLo alcohol products, resulting in ineffective regulation of such products because one category can be sold as the other. The delimitations are also critical for international commerce and for taxation purposes.

Another subcategory of no alcohol is “zero” or “0.0% vol.”, for which, the analytical limits would be set at 0.04% vol., while 0.05% vol. would be rounded off to 0.1% vol. From a biochemical standpoint, this category is often more a marketing-driven category because ethanol at levels of 0.5% vol. already appears virtually safe. However, some groups of consumers, such as those of the Islamic religion, may prefer 0.0% vol. for ethical reasons. According to Alzeer and Abou Hadeed, any ethanol concentration as low as 0.1% vol. that is prepared with the intention of being used as a beverage drink is considered non-Halal [16].

## 4. Artisanal No and Low (NoLo) Alcohol Products

Fermented alcoholic beverages with less ethanol have been made in cultures throughout history. For example, it is generally known that tree saps such as palm sap, and birch sap wine have been produced and consumed in European nations [1]. Since the saps contain very little sugar, the beverages made from them should have a very low alcoholic strength (about 0.5% vol.). A variable alcohol content has been observed for boza, a lactic acid fermented drink, as a result of different traditional recipes. Boza has been found to contain alcohol with an alcoholic strength of less than 1% vol. in Turkey and up to 7% vol. in Egypt, most likely due to microorganisms active during the fermentation process [1]. Brazilian kombuchas are divided into two categories: “low alcohol” (less than 0.5% vol.) and “alcoholic” (over 0.5% vol. but under 8% vol.) [1]. Table 2 gives the approximate alcoholic strength of some artisanal products [17].

Although studies have shown that these beverages have low alcoholic strength, they can suffer from poor quality, as some can be contaminated with substances such as mycotoxins [20] and heavy metals [21]. Additionally, because of their low alcohol content, they are more susceptible to microbial contamination that, in turn affects their shelf life. Consequently, there is a compelling need for in-depth characterisation of the contents of artisanal NoLo alcohol products, including microbial ecology, to guide the development of regulations on quality monitoring of the beverages.

## 5. Alcohol-Flavoured No-Alcohol Products

Alcohol-flavoured, non-alcoholic beverages refer to a category of products whereby minimal quantities of alcohols or alcohol-resembling flavours are added to improve the sensory attributes of the beverages, intending to imitate alcohol flavour or the flavour composite of a specific alcoholic beverage (e.g., a certain cocktail). This is in contrast to conventional NoLo alcohol products that are based on their alcoholic counterparts, but are produced by either alcoholic fermentation, which is stopped at a certain low level, or the conventional products are dealcoholized afterwards. 

The category of alcohol-flavoured, no-alcohol products generally refers to alcohol-free spirits, which are mostly based on flavours that try to imitate the original product. When considering alcoholic beverage-resembling flavours in this context, this goes much beyond ethanol to category-characterising flavours that may encompass a broad range of various volatiles.

Although the alcohol content in these alcohol-flavoured beverages is typically below 0.1% vol., some residual alcohol may be expected from the use of food flavours, which are often dissolved and commercialised in alcoholic solutions. Figure 1 provides a classification of NoLo alcohol products that includes artisanal and alcohol-flavoured non-alcoholic beverages.

## 6. Production Techniques of No or Low (NoLo) Alcohol Beers

No or low alcohol products are produced as non-fermentative or fermentative beverages. In the fermentative type, alcohol is either avoided or later removed from the liquid using physical techniques such as heating (distillation), dialysis, reverse osmosis, or biological methods. Biological methods involve either the use of a low alcohol producing yeast strain and/or halting fermentation by heating the wort [10]. Excellent reviews of the production of alcohol-free beverages have been published [10,12,22,23,24].

## 7. Quality Control Aspects of No and Low (NoLo) Alcohol Products

### 7.1. Microbial Ecology

NoLo alcohol products are more susceptible to microbial contamination due to their low alcoholic strength and the presence of fermentable sugars [25,26]. A large number of lactic acid bacteria, coliforms, moulds, and yeasts cause spoilage, as they can use the carbohydrate content for fermentation processes producing undesirable changes in them. Acids, alcohols, and diacetyl are examples of fermentation by-products that impact the organoleptic quality of the beverages. This deterioration is also accelerated by the sugar used as a sweetener [25]. NoLo alcoholproducts are, therefore, typically either heat-sterilized or aseptically packaged and should have stability similar to that of other alcohol-free beverages, such as fruit juices or caffeinated soft drinks.

### 7.2. Alcoholic Strength and Chemosensory Perception 

As a precursor to flavour-active esters, ethanol enhances some flavours, such as those that result in a sweet taste. In addition to warming the olfactory tissues, ethanol is recognized to have a significant influence on the development of the distinctive background flavour of beer [9]. In NoLo alcohol products, a partial loss of flavour is unavoidable because ethanol is eliminated using various dealcoholisation techniques [9,27]. Ethanol has a considerable impact on the flavour release of alcoholic beverages and subsequent chemosensory perception. Discrimination of alcoholic strength by taste has been shown to be only partially possible in a range of intermediate alcoholic strengths and is not possible at higher concentrations [28,29]. Alcoholic beverages′ scent, taste, and mouthfeel might be perceived differently by consumers depending on the amount of ethanol in the beverage [30]. The perception of worty taste is diminished due to ethanol′s increased aldehyde retention. Conventional beers retain aldehydes at a rate of 32–39% as compared with 8–12% in alcohol-free beers [10].

### 7.3. Higher Alcohols

Higher alcohols, also known as fusel oils, occur in fermented alcoholic beverages. The concentrations of these alcohols are dependent on the efficiency of amino acid uptake and sugar utilization by the yeast [31]. Higher alcohols are classified into aliphatic and aromatic alcohols. The main aliphatic higher alcohols are 1-propanol, isobutanol, amyl alcohol, and isoamyl alcohol, while the main aromatic higher alcohols are 2-phenylethanol, tyrosol, and tryptophol [32].

1-Propanol, isobutanol, and isoamyl alcohols make up most of the aliphatic higher alcohols that contribute to the aroma and warmth of beer [9,32]. 1-Propanol and 2-methylpropanol can contribute to the “rough” flavours and harshness of beer, while amyl alcohols contribute to the “fruity” qualities [33]. When the isobutanol concentration exceeds 20% of the combined concentration of three alcohols (1-propanol, isobutanol, and amyl alcohol), it has a negative impact on beer quality [34]. In contrast to the disagreeable odours created by tyrosol and tryptophol, the aromatic alcohol 2-phenylethanol imparts “sweet” or “rose” qualities into beer [32,33].

In particular, spirit-similar beverages made with added flavours are currently not expected to contain higher alcohols. However, flavour mixes of the higher alcohols representing a specific type of spirit might be conceivable. Fusel alcohols in NoLo alcohol products, if they do not exceed the levels currently reported to naturally occur in their alcohol-containing counterparts, are likely not to pose an inherent health risk to consumers [35]. 

However, as mentioned in Section 2 above, higher alcohols may contribute to alcoholic strength; therefore, at least for “zero” or “no” alcohol products, the product formulation must also exclude these components. For low-alcohol products, the level of alcoholic strength that includes both ethanol and higher alcohols must stay below the threshold of the respective jurisdiction.

Although there are some quality characteristics of alcohol-free beer that are similar to those of conventional beer, such as pH, colour, and turbidity, there are notable changes in the olfactory components, such as alcohols, aldehydes, and esters. Unfortunately, the absence of aroma components may result in products that consumers find to be sensory unacceptable; however, recently, the sensory quality in the category has improved considerably, for example, by blending beer from stopped fermentation (which is rather sweet) with beer from distillation-type dealcoholisation.

### 7.4. Sugar Content

Alcoholic beverages can contain significant amounts of sugar [36]. Beyond alcohol-related health risks, other considerations of health policy are the health risks caused by excessive sugar consumption. Sometimes, substitution of alcohol with sugars may occur in NoLo alcohol products. (e.g., in the category of alcohol-free beers, manufactured using stopped fermentation, or in some alcohol-free spirits on the market, which are mostly coloured and flavoured sugar-water [28]). The amount of total carbohydrates in alcohol-free beers is typically about twice that of alcohol-containing beers. The net reduction in calories may still be significant. This can be explained by the fact that alcohol contains almost twice as many calories per gram as carbohydrates and proteins [8]. For example, in a sample of more than 1200 German beers, the average energy was reduced by almost half from 181 kcal/100 mL (alcohol-containing beers) to 98 kcal/100 mL (alcohol-free beers) [37]. Nevertheless, the net public health effects of alcohol are much higher than those of sugar; therefore, some substitution of alcohol with sugar would probably still be advantageous from a public health standpoint.

However, higher sugar levels could discourage some consumers from switching to NoLo alcohol products. Sugar reduction strategies are necessary to influence consumer choices. It remains to be seen whether consumers will willingly choose high sugar content over high alcohol content. There is clearly much room for product development, including sugar-free NoLo alcohol products.

Another aspect of the balance between sugar and alcohol is the isotonicity of the beverages. Although alcohol-containing beers generally are in the hypertonic osmolality range, many alcohol-free beers fall into the preferable isotonic range, making the products interesting, for example, to restore the strength of people who perform strong physical exercise [38].

### 7.5. Other Compounds of Public Health Concern

In addition to ethanol, other constituents of alcoholic beverages pose potential health risks to consumers. The contamination of raw materials with mycotoxins [20,21,39], such as aflatoxins and ochratoxin A, is another potential concern for the derived NoLo alcohol products, especially in temperate and tropical regions. However, this can be mitigated by adapting good agricultural practices to keep contaminants as low as reasonably achievable. Similarly, heavy metals such as iron, lead, copper, or zinc can occur in NoLo alcohol products. Other volatile compounds, such as acetaldehyde and methanol, are potential contaminants worthy of monitoring in alcoholic beverages. Therefore, the appropriate preventive measures already existing for conventional alcoholic beverages can help to control and minimize the occurrence of these contaminants.

## 8. Quality Control Technologies of No and Low (NoLo) Alcohol Products

Globally, there are regulations for various compounds in beverages and foods; therefore, there is a need for methods to control levels. The main objective of fit-for-purpose analytical technologies would, firstly, be to determine the alcoholic strength, as this is the main marketing claim on the beverages. Additionally, it would be worthwhile to determine other constituents that impact consumers health and influence consumer choices.

Currently, there are no available methods that are specifically related to NoLo alcohol products, but typically all methods available for alcoholic and alcohol-free beverages can be used. The methods to determine alcoholic strength must be tested, if the sensitivity is sufficient for the needed lower levels, but in the authors′ experiences regarding official alcohol control, densimetric methods (such as oscillation-type electronic densimetry) for determining the EU requirements can determine alcoholic strength with one decimal place [8] as low as 0.0% vol. (which mathematically would be less than 0.04% vol.). Therefore, the limit of detection of the method should not be higher than this level (i.e., 0.04% vol. for controlling a level of 0.0% vol).

In the authors’ experience, more modern methods than densimetry, such as nuclear magnetic resonance (NMR) spectroscopy, can also be used for non-destructive determination of constituents of NoLo alcohol products, since they have been previously used for identification and quantitation of multiple compounds. In a study, 36 different ingredients (i.e., sugars, flavourings, sweeteners, organic acids, alcohols, and vitamins) were successfully identified and quantified in alcohol-free beverages [36,40]. Moreover, since NoLo alcohol products are known to have a high sugar content, proton NMR could provide a rapid and efficient technique to monitor the sugars, namely fructose, glucose, and sucrose [36]. 

Other mainstream techniques for the analysis of beverages are still useful. The techniques range from simple pycnometry to advanced chromatography, sensory analysis, and spectroscopy [36,41,42,43,44,45,46]. The applications can be customised to analyse physicochemical and sensory attributes as necessary. There are excellent reviews on analytical techniques applied to the analysis of alcohol and non-alcoholic beverages [47]. 

## 9. Labelling of No and Low (NoLo) Alcohol Products

Labelling of NoLo alcohol products is discussed in more detail in a separate Special Issue article [48]. In brief, information on the ingredients is necessary for health-conscious consumers to choose the beverages. A declaration of ingredients and nutrients is crucial to provide consumer information about the product. For NoLo alcohol products, it is critical that producers provide consistent nutritional information, including details on the content of ethanol, total carbohydrates, and any other additives that impact consumer health. This would also refer to other alcohols, in addition to ethanol, that may cause relapse due to alcohol-related stimuli in abstinent persons with former alcohol dependence [49]. Currently, there is no specific legislation for NoLo alcohol products, apart from the general food information requirements necessary for any category of food [8]. 

The fundamental justification for including NoLo alcohol products in global legislation is that they are thought to offer considerable health benefits if alcohol consumption is partially or fully substituted. Nutritional labelling has been suggested to be a strategy to help people choose foods that are advantageous for them to buy and eat [50]. 

There may be a comparable level of concern about the lack of adequate health information on NoLo alcohol products, given the typical lack of appropriate nutritional labelling on alcoholic beverages. Research has even proven that beverages with an alcoholic strength of 0.5% vol. or less have no physiological effects on the body and cannot cause intoxication [11]. However, low-strength products of up to 1.2% vol. may pose hazards, particularly for pregnant women and those with underlying medical problems, or persons in abstinent remission of alcohol use disorders [51,52,53,54]. Such health implications should be stated on the label.

## 10. Discussion

Although NoLo alcohol products ostensibly provide a solution to alcohol-related health risks in society even if they contain only trace levels of alcohol, there are still problems associated with deceptive marketing, advertising, underage drinking, and adverse health effects that require appropriate regulation. The aim of regulation should be to ensure that the market for NoLo alcohol products develops in a way that promotes public health rather than harms it. Another aspect that merits research is the potential for individuals to consume the same volume of alcohol (the number of drinks consumed) at a lower strength. Regulating these products may become increasingly difficult as the line between soft drinks and alcohol replacement beverages blurs. For instance, beverage producers might claim that soft drinks are alternatives to alcohol in an effort to avoid sugar taxes imposed on certain categories of non-alcoholic beverages. In addition to the negative effects of alcohol, the harmful effects of sugar consumption should also be considered when developing holistic health policies. There is a need to conduct additional research on the associations between the various NoLo descriptors, consumer behaviours, and health consequences. There is also the interesting possibility of marketing NoLo alcohol products as specialty products that may even have additional health benefits (such as isotonicity) without alcohol-related health risks. For health reasons or to prevent alcohol’s aftereffects, such as driving impairment, hangovers, and lethargy, most NoLo alcohol product consumers prefer NoLo alcohol alternatives for specific times rather than abstaining from alcohol entirely.

Alcohol consumption is currently associated with a high risk of morbidity and mortality [29]. The strategy of reducing the alcoholic strength of beverages while maintaining consumer sensorial preferences could mitigate the harmful effects of alcohol consumption in society. A reduction in the potency of alcohol products as a measure of reducing overall consumption and alcohol-related harm is encouraged. 

## 11. Conclusions

Evidently, there is no clear-cut classification of low alcoholic content and non-alcoholic beverages. There is a compelling need for unambiguous standards and definitions for no- and low-alcohol products spanning the different categories of beers, wines, and spirits [55].

Taking into consideration the available evidence, the authors currently suggest the categories defined in Table 3 for NoLo alcohol products. While the “reduced/light” category is already defined in Regulation (EC) No. 1924/2006 [56], the other categories should be included in the regulation similar to the “low-fat” and “fat-free” class that are already included. According to the authors’ belief, the mandatory labelling of alcoholic strength should be lowered to 0.5% vol., while labelling below 0.5% vol. appears incommensurate as this would also meet other food groups and appears to be unnecessary from a biochemical standpoint, because this level appears virtually safe. This would be a good compromise between consumer and industry acceptability and public health. Even more preferably would be an internationally aligned standard, for example, on the Codex Alimentarius international food standard level by the FAO and the WHO. 

The authors recommend strict implementation and enforcement of labelling requirements for all alcoholic beverages to allow informed decision making by consumers. Additionally, an increase in the availability of no and low alcohol products could significantly contribute to a reduction in harmful alcohol use.

## Figures and Tables

**Figure 1 nutrients-14-03873-f001:**
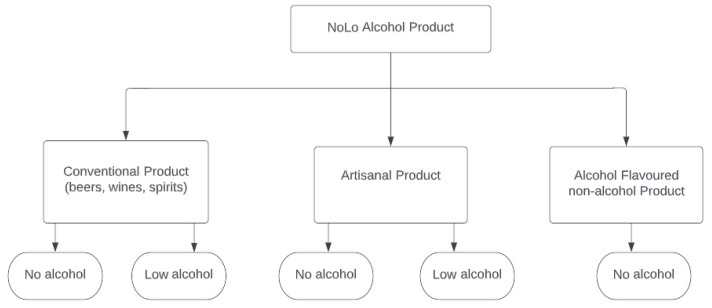
Classification of no and low (NoLo) alcohol products.

**Table 1 nutrients-14-03873-t001:** Conventions of ethanol/alcoholic strength in no and low (NoLo) alcohol products for different countries (Adapted from [14,15]).

Country	Ethanol/Alcoholic Strength (% vol.)	
	No-Alcohol Products	Low-Alcohol Products
Austria	≤0.5	ND
Belgium	≤0.5	ND
Croatia	≤0.5	ND
Cyprus	≤0.5	ND
Czech Republic	≤0.5	ND
Denmark	≤0.5	0.5–2.8
Finland	≤2.8	2.8–3.7
France	≤1.2	ND
Germany	≤0.5	ND
Hungary	≤0.5	ND
Iceland	≤2.25	ND
Italy	≤1.2	ND
Netherlands	≤0.1	ND
Norway	≤0.7	0.7–2.75
Portugal	≤0.5	ND
Slovenia	≤1.2	ND
Spain	≤1.0	ND
Sweden	≤0.5	ND
Australia	≤0.5	ND
Canada	≤0.4	ND
United Kingdom	≤0.05	0.05–1.2
USA	≤0.5	ND
China	≤0.5	ND
Turkey	≤0.5	ND
Japan	≤1.0	ND
India	≤0.5	ND
South Africa	≤0.5	ND
Nigeria	≤0.5	0.5–1.0
Kenya	ND	ND

ND, not defined.

**Table 2 nutrients-14-03873-t002:** Alcoholic strength of selected fermented products that may contain no or low levels of alcohol (Adapted from [18,19]).

Name (Country/Region)	Alcoholic Strength (% vol.)
Kefir (Russia, Eastern Europe)	<2
Boza (Egypt, Turkey, Bulgaria, Albania, and Romania)	Typically < 1.5 Turkish 0.03–0.39 (*w*/*v*)
Kvass (Russia)	≤1
Hardaliye (Turkey)	0.28–0.59
Kombucha (Brazil)	≤0.5
Socata (Romania)	≤0.5
Ginger beer (UK)	≤0.5
Birch beer (USA, Canada)	≤0.5
Sima (Finland)	≤0.5
Chicha de jora (Argentina, Ecuador, Peru)	1–3
Sekete (Nigeria)	1–3
Omalovu (Namibia)	0.2–4.1
Bili bili (Chad)	1–8
Dolo (Burkina Faso, Benin, Rwanda)	1–5

**Table 3 nutrients-14-03873-t003:** Suggested definitions for no and low (NoLo) alcohol products in the European Union.

Category for Nutritional Claim	Alcoholic Strength Range	Labelling of Alcoholic Strength
Zero alcohol or 0.0% vol.	<0.04% vol.	0.0% vol. ^1^
No alcohol or alcohol free	<0.5% vol.	Not mandatory
Low alcohol	0.5–1.2% vol.	Suggested to be mandatory ^2^
Reduced alcohol or light alcohol	Reduction in content of at least 30% as compared with a similar product ^3^	Mandatory depending on actual level

^1^ The 0.0% vol. or zero alcohol category could be labelled on a voluntary basis. But if advertising claims are made in this regard, labelling of alcoholic strength should be made mandatory. ^2^ The authors suggest reducing the mandatory labelling level from 1.2% vol. [8] to 0.5% vol. Until then, voluntary labelling is currently encouraged. ^3^ Requirement for reduced nutrient according to Regulation (EC) No. 1924/2006 [56]. For example, a reduced alcohol Pilsner beer of a certain brand must have 30% reduced alcohol as compared with the standard-type Pilsner of the same brand (i.e., a beer with an alcoholic strength of 5.5% vol. would have to be reduced to an alcoholic strength of 3.8% vol.).

## Data Availability

No new data were created or analyzed in this study. Data sharing is not applicable to this article.

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
