# Peer review of "Defining No and Low (NoLo) Alcohol Products"

_nutrients, 2022, doi:10.3390/nu14183873_

Round 1

Reviewer 1 Report

First of all I would like to congratulate the authors for the work done.

They have achieved a complete vision of the problem and very accessible to the reader.

I would like to be able to make two considerations that can be enriching for this research:

In the first place, approach from a conceptual perspective the legality of consumption in the different countries. Extremely conditioning factor when consuming.

Secondly, the interest in health, which concerns the consumption of certain alcoholic beverages, such as alcohol in the Mediterranean diet

Author Response

First of all I would like to congratulate the authors for the work done.

They have achieved a complete vision of the problem and very accessible to the reader.

REPLY: Thank you very much!

I would like to be able to make two considerations that can be enriching for this research:

In the first place, approach from a conceptual perspective the legality of consumption in the different countries. Extremely conditioning factor when consuming.

REPLY: NoLo should be legal in most countries, at least below 1.2% vol, which is typically not seen as alcoholic beverages, perhaps apart from Muslim countries, where 0,0 could be demanded.

Secondly, the interest in health, which concerns the consumption of certain alcoholic beverages, such as alcohol in the Mediterranean diet.

REPLY: This would probably be a separate review. The Mediterranean or French parodox is typically believed to be predominantly based on confounding.

There exists a plethora of literature on the perceived link between intake of a Mediterranean  diet which typically is high in fruit, vegetables, and grains, and also includes one to two drinks per day and longevity. Whether one to two drinks adversely affect cancer incidence in the presence of a Mediterranean diet has not been fully explored. With the evidence currently available, we believe the cardioprotective components in the Mediterranean diet may not entirely mitigate against the more detrimental effects of alcohol.

We do not believe that there is currently much evidence that no or lo alcoholic beverages might be associated with positive health effects, e.g. protective or antioxidant effects from polyphenols without the detrimental effects of alcohol. This might be an interesting topic for future investigations, however.

Reviewer 2 Report

The nutrients-1882281 is well-summarized no or low alcohol products worldwide; however, the nutrients-1882281 may be out of scope for the Nutrients. If the nutrients-1882281 is acceptable, figure 1 should be updated in a more professional way.

Author Response

REPLY: This is an invited review article for a special issue in Nutrients on no and low alcoholic beverages. The editor invited us specifically asking for a paper on „Defining No and Low Alcohol (NoLo) Products“. We therefore believe that the paper is not out of scope for this special issue on Nutrients. Besides that, ethanol could be seen as a special form of nutrient (or anti-nutrient). For example, in the European Union, the energy (calory) labelling has to include the calories of alcohol (7 kcal/g).

We have redrawn Figure 1 as suggested by the reviewer.

Reviewer 3 Report

The  manuscript by Okaru and Lachenmeier is  an interesting review on the very relevant - but poorly explored - issue of no alcohol and low alcohol beverages. I found it generally well-written, well-organized, informative and clear. However, I am concerned about a few but relevant issues:

- acute and chronic harmful effects of alcohol - and related dosages - should be briefly presented;

- references should be provided for the statement at lines 131-132.

In particular, please note that alcohol assumption during pregnancy is especially dangerous for the fetus and no alcohol dose is considered safe (Centers for Disease Control and Prevention). In particular, although high doses and risky patterns of alcohol consumption can have a high impact on cognitive functions, affective dimension and future alcohol vulnerability of the offspring, also low risk alcohol intake can affect the mother-infant dyad (see: doi: 10.3389/fpsyt.2018.00150 , doi: 10.3389/fnbeh.2016.00031 , doi: 10.3389/fnbeh.2020.583122 , doi: 10.1515/revneuro-2017-0052 ).

Minor point:  the abstract has to be rewritten, because I find the introduction (lines 9-16) too long, while the methods/results paragraphs are not present

Author Response

The  manuscript by Okaru and Lachenmeier is  an interesting review on the very relevant - but poorly explored - issue of no alcohol and low alcohol beverages. I found it generally well-written, well-organized, informative and clear. However, I am concerned about a few but relevant issues:

- acute and chronic harmful effects of alcohol - and related dosages - should be briefly presented;

REPLY: We have include a brief paragraph on the acute and chronic effects of alcohol consumption

- references should be provided for the statement at lines 131-132.

In particular, please note that alcohol assumption during pregnancy is especially dangerous for the fetus and no alcohol dose is considered safe (Centers for Disease Control and Prevention). In particular, although high doses and risky patterns of alcohol consumption can have a high impact on cognitive functions, affective dimension and future alcohol vulnerability of the offspring, also low risk alcohol intake can affect the mother-infant dyad (see: doi: 10.3389/fpsyt.2018.00150 , doi: 10.3389/fnbeh.2016.00031 , doi: 10.3389/fnbeh.2020.583122 , doi: 10.1515/revneuro-2017-0052 ).

REPLY: We have added references. Thank you for the enriching comments.

Minor point:  the abstract has to be rewritten, because I find the introduction (lines 9-16) too long, while the methods/results paragraphs are not present

REPLY: We have made minor changes/restructuring to accomodate the reviewer’s comments.

Round 2

Reviewer 2 Report

No comment.

Reviewer 3 Report

The authors have addressed the main points I raised.